# Feasibility of Adjuvant Treatment with Abemaciclib—Real-World Data from a Large German Breast Center

**DOI:** 10.3390/jpm12030382

**Published:** 2022-03-02

**Authors:** Dominik Dannehl, Lea L. Volmer, Martin Weiss, Sabine Matovina, Eva-Maria Grischke, Ernst Oberlechner, Anna Seller, Christina B. Walter, Markus Hahn, Tobias Engler, Sara Y. Brucker, Andreas D. Hartkopf

**Affiliations:** 1Department of Women’s Health, Tuebingen University, 72076 Tuebingen, Germany; lea-louise.volmer@med.uni-tuebingen.de (L.L.V.); martin.weiss@med.uni-tuebingen.de (M.W.); sabine.matovina@med.uni-tuebingen.de (S.M.); eva-maria.grischke@med.uni-tuebingen.de (E.-M.G.); ernst.oberlechner@med.uni-tuebingen.de (E.O.); anna.seller@med.uni-tuebingen.de (A.S.); christina-barbara.walter@med.uni-tuebingen.de (C.B.W.); markus.hahn@med.uni-tuebingen.de (M.H.); tobias.engler@med.uni-tuebingen.de (T.E.); sara.brucker@med.uni-tuebingen.de (S.Y.B.); andreas.hartkopf@uni-tuebingen.de (A.D.H.); 2Research Institute for Women’s Health, Tuebingen University, 72076 Tuebingen, Germany; 3Department of Gynecology and Obstetrics, University Hospital of Ulm, 89081 Ulm, Germany

**Keywords:** breast cancer, oncology, CDK 4/6, systemic therapy, monarchE, abemaciclib

## Abstract

Abemaciclib significantly improves invasive disease-free survival when combined with endocrine therapy in clinical high-risk patients with HR+/Her2− early breast cancer (eBC). The objective of the following study was to model how many patients with eBC would be available for adjuvant treatment with abemaciclib in a real-world setting. Patients that underwent complete surgical treatment for eBC between January 2018 and December 2020 in a large single-center university hospital in Germany were eligible. Descriptive statistics were used to describe the patient population that could benefit from abemaciclib according to the inclusion criteria of monarchE. Of 1474 patients with eBC, 1121 (76.1%) had a HR+/Her2− subtype. Of these, 217 (19.4%) fulfilled the monarchE inclusion criteria. Within patients that fulfilled the monarchE inclusion criteria, 48.9% received no adjuvant or neoadjuvant chemotherapy. Thus, in a real-world situation, fewer patients will be pretreated with chemotherapy than was the case in monarchE. Breast care units are facing a significant patient burden, since the 2-year abemaciclib therapy requires regular monitoring of toxicities. Specific care concepts to strengthen therapy adherence as well as further studies to deescalate adjuvant systemic treatment and individualize CDK 4/6 inhibitor therapy are therefore needed.

## 1. Introduction

In the last few years, advances in the personalized treatment of breast cancer were able to extend the prognosis of patients [1]. The use of cyclin-dependent kinase 4/6 (CDK 4/6) inhibitors in hormone receptor positive (HR+) and human epidermal growth factor receptor 2 negative (Her2−) advanced or metastatic breast cancer improves progression-free and overall survival and is now the standard of care [2,3,4,5,6,7,8]. Thus, several studies have attempted to adapt these effective therapeutic approaches to earlier disease stages (PenelopeB, NCT01864746, palbociclib [9]; monarchE, NCT03155997, abemaciclib [10]; NATALEE, NCT03701334, ribociclib [11]; Pallas, NCT02513394, palbociclib [12]). In particular, the monarchE trial (NCT03155997) investigated the efficacy and safety of the CDK 4/6 inhibitor abemaciclib in combination with endocrine therapy in patients with lymph-node-positive HR+/Her2− early breast cancer (eBC). These were patients with clinicopathological high-risk features who underwent surgery and had completed radiotherapy and/or (neo)adjuvant chemotherapy [10,13]. After 2 years of abemaciclib (2 × 150 mg/d) in combination with standard endocrine therapy, there was a statistically significant improvement of invasive disease-free survival (iDFS) with a hazard ratio and 95% confidence interval of 0.71 and 0.58–0.87, respectively, which recently led to approval by the U.S. Food and Drug Administration (FDA) [13,14]. Although monarchE also included patients with a Ki67 < 20% but high risk clinicopathological factors (involvement of ≥3 lymph nodes, histologic grade 3 or tumor size ≥ 5 cm), the FDA approved abemaciclib in combination with endocrine therapy for node-positive, early breast cancer at high risk of recurrence only if the Ki67 is at least 20%. Presuming approval also by the European Medicine Agency (EMA), the aim of this study was to model how many patients with eBC would be eligible for abemaciclib treatment using the inclusion criteria of monarchE and clinical data from a large single-center university hospital in Germany. Since clinical trials are conducted under controlled conditions; this study characterizes the patient population benefiting from abemaciclib in a real-world setting.

## 2. Materials and Methods

All patients included in this retrospective analysis were treated for eBC at the Department of Women’s Health at Tuebingen University Hospital, Germany. The study was conducted according to the guidelines of the Declaration of Helsinki and approved by the Ethics Committee of Tuebingen University Hospital (protocol code 075/2022BO2). Only patients (female and male) who underwent complete (R0) surgical treatment between January 2018 and December 2020 were eligible. Exclusion criteria were distant metastatic disease, lack of proliferation marker Ki67, and missing information on lymph node involvement. If patients were diagnosed with bilateral breast cancer, the tumor with the worse prognosis was included in the study (Figure 1). Tumors were counted as HR+ if they had a positive estrogen receptor (ER) and/or a positive progesterone receptor (PR) expression according to immunohistochemistry (≥1% for ER, ≥10% for PR). The Her2 status was assessed to local standards by using the HERCEPT test (DAKO, Glostrup, Denmark). Expression of Her2 was scored on a 0 to +3 scale. Tumors with a score of +3 were considered Her2-positive. In case of a score of +2, Her2 amplification was determined by fluorescence in situ hybridization using the Pathvysion^®^ Kit (Vysis, Downers Grove, IL, USA). Ki67 was assessed using the M7240 monoclonal mouse anti-human Ki-67 antibody MIB-1 (Agilent Dako, Santa Clara, CA, USA).

To assess which patients would be eligible for abemaciclib treatment, the inclusion criteria of the monarchE trial were applied: HR+/Her2− lymph-node-positive eBC with either proliferation marker Ki67 ≥ 20% or Ki67 < 20% and (i) at least 4 pathologic lymph nodes (N2), (ii) histologic grade 3 (G3), or (iii) tumor size of at least 50 mm (T3) [10].

Data processing and statistical analysis were performed using Jupyter Notebook (Version 6.3.0, Project Jupyter, open-access and community developed) on Anaconda (Version 3.0, Anaconda Inc., Austin, TX, USA) with the Python extension packages pandas (Version 1.4.1, open-access and community developed) and numeric Python (Version 1.22.2, open-access and community developed). Lucid^®^ (Lucid Software Inc., South Jordan, UT, USA) was used for designing flow charts and data visualization.

## 3. Results

Of 2257 patients who underwent surgery for eBC at the Department of Women’s Health at the University of Tuebingen, Germany, between 2018 and 2020, 783 were not included in the analysis (Figure 1). Of the remaining 1474 patients, the mean age was 58.8 ± 12.9 years and the most common tumor biology was HR+/Her2− (76.1%), followed by Her2+ (14.7%) and triple negative (9.2%). Since only patients with HR+/Her2− early breast cancer are eligible for treatment with abemaciclib, the following section will focus on HR+/Her2− patients only (Table 1). Patient characteristics of the whole study population can be reviewed in Table A1.

Among 1121 HR+/Her2− breast cancer patients, the most common histology was non-special type (75.8%). The most common histologic grading was G2 (65.6%). Of all tumors, 91.5% were T1-2 and 29.7% of all HR+/Her2− patients had involved pathologic lymph nodes. Of all tumors, 40.5% exhibited a Ki67 ≥ 20%. Of all HR+/Her2− patients, 76.6% did not undergo chemotherapy. A total of 262 patients (23.4%) underwent systemic therapy, 62 (5.5%) were treated using neoadjuvant chemotherapy, and 200 (17.8%) underwent adjuvant chemotherapy (Table 1).

As displayed in Figure 2, 217 (19.4% of all HR+/Her2− patients) fulfilled the inclusion criteria of the monarchE trial. Among these, 174 (15.5% of all HR+/Her2− patients) exhibited a tumor with Ki67 ≥ 20% (80.2% of the patients fulfilling the monarchE inclusion criteria).

Table 2 shows the characteristics of 217 patients who fulfilled the inclusion criteria of monarchE. The mean age of these patients was 60.2 ± 14 years, 68.7% were postmenopausal, 30.0% were premenopausal, and 1.4% were male. The most common tumor subtype was non-special type (80.7%). The tumor stage was T1-2 in 78.8% of patients. All included patients had pathologic lymph node involvement either at the beginning of neoadjuvant chemotherapy or after surgery prior to adjuvant chemotherapy. In total, 26 patients (12%) received neoadjuvant and 85 patients (39.2%) adjuvant chemotherapy.

## 4. Discussion

The CDK 4/6 inhibitor abemaciclib is the first of its kind to significantly prolong iDFS in patients with HR+/Her2− eBC and clinicopathological high-risk features [10,13]. Evaluating data from a large single breast cancer center that is, in our opinion, representative of the overall population of patients with eBC at least in Germany, we found that 19.4% of all HR +/Her2− patients (i.e., 14.7% of all patients with eBC, regardless of the tumor biology) could benefit from this new treatment.

In contrast to monarchE, other studies investigating CDK 4/6 inhibition with palbociclib for the treatment of eBC showed no statistically significant benefit [9,12]. Next to chemical differences between abemaciclib and palbociclib, differences in the study population are likely to explain the different results of these trials as compared to monarchE. Ribociclib, another CDK 4/6 inhibitor, is currently under investigation in the NATALEE trial [11]. In addition, it is too early to draw any conclusions on the long-term impact of adjuvant CDK 4/6 inhibition in high-risk eBC. As the biology of early relapse may be different from that of late relapse, prolonged follow-up data are critical, and currently there is no proven impact of adjuvant abemaciclib on overall survival [15].

Recently published data from an additional follow-up of monarchE revealed that patients in the Ki67 high cohort displayed a 3-year iDFS of 86.1% when treated with abemaciclib compared to 79.0% with placebo [13]. A similar effect was observed in the Ki67 low cohort that reached a 3-year iDFS in 91.7% with abemaciclib and 87.2% with placebo. Nevertheless, FDA approval for abemaciclib did not include patients with a Ki67 < 20%. In our study, 15.5% of all HR+/Her2− patients (i.e., 11.8% of all patients with eBC, regardless of tumor biology) fulfilled the FDA label. Approval from the EMA is pending.

An important factor for therapy decision in the monarchE trial was the Ki67 proliferation index [10,14]. Yet, as shown by different trials and a recent publication of the International Ki67 in Breast Cancer Working Group (IKWG), the analytical validity and clinical utility of Ki67 is limited due to technical procedures and the diverging interrater reliability [16]. However, the IKWG consensus states that a Ki67 of 5% or less or of 30% or more can be used to identify patients who would benefit from adjuvant chemotherapy [16]. Due to the limitations and the heterogeneity of Ki67 assessment, other diagnostic tools could discriminate between patients that benefit from an adjuvant treatment with a CDK 4/6 inhibitor and those who do not. In HR+/Her2− node-negative and node-positive eBC gene expression assays have proven to identify patients that benefit from adjuvant chemotherapy (OncotypeDx, MammaPrint, Endopredict, PAM50) [17,18,19,20,21,22]. The NATALEE trial is currently investigating the role of ribociclib for adjuvant treatment of HR+/Her2− node-positive eBC, and allows, next to Ki67, the use of the aforementioned gene expression assays for the identification of high-risk patients [11].

In our real-world cohort, nearly half of the patients that would be eligible for adjuvant abemaciclib treatment received no adjuvant or neoadjuvant chemotherapy. This is in line with data from Pivot et. al. [23], who performed a real-world study on high-risk patients with early breast cancer in France using data from 412 patients reported by different physicians, but considerably lower than the trial population of monarchE, where 95.4% of all patients had received adjuvant or neoadjuvant chemotherapy [10,13]. One explanation may be differences in the trial population, with more patients being postmenopausal and more patients having less than four involved lymph nodes in our population than in monarchE. Additionally, in a real-world situation, more patients may not be willing to undergo chemotherapy or have contraindications. Contrary to these findings, more patients in our populations had a Ki67 of at least 20%; however, the use of gene signatures could have spared some of these patients from using chemotherapy. The monarchE study cannot answer the question of whether the use of CDK 4/6 inhibitors can avoid the need for adjuvant chemotherapy, which is already the case in the first-line treatment of metastatic breast cancer [24]. Studies on the de-escalation of adjuvant systemic therapy using modern therapeutic strategies are therefore particularly important. This question is currently being investigated in the ADAPTcycle trial (NCT04055493), in which patients at intermediate risk of relapse are randomized to adjuvant ribociclib or chemotherapy [25].

This study highlights the importance of characterizing the patient population that could benefit from abemaciclib in a real-world situation. The results of this study are of great importance for estimating the target group and thereby the cost-effectiveness of new therapeutic approaches. However, these findings should be evaluated carefully since we conducted a single-center analysis that might not reflect the standard of treatment decisions in Germany. Especially with regard to increasing treatment possibilities for breast cancer, not all putative drug combinations can be evaluated in ‘classical’ clinical trials. Therefore, multicentric registers are needed to precisely collect information about treatment process, clinicopathological risk factors, molecular data, and patient outcome [5,26].

In conclusion, up to 20% of all patients with HR+/Her2− eBC are eligible for abemaciclib treatment. However, the decision whether to choose intensive adjuvant treatment not only depends on histopathologic factors but also on patients-related factors like comorbidities, toxicity, and individual preferences. In a real-world situation, fewer patients will be pretreated with chemotherapy than was the case in monarchE. It is also conceivable that precisely because a CDK 4/6 inhibitor is now available in the adjuvant setting, some physicians and patients may decide against the use of chemotherapy. However, regular monitoring of toxicities is of utmost importance since patients that received the CDK 4/6 inhibitor palbociclib discontinued treatment in 27.2% of cases due to adverse effects in the PALLAS trial [27]. Hence, the implementation of care plans such as specially trained nurses is needed. Data from real-world registries and from future prospective trials are required to safely deescalate adjuvant systemic treatment and to define the patient population that is most likely to benefit from CDK 4/6 inhibition.

## Figures and Tables

**Figure 1 jpm-12-00382-f001:**
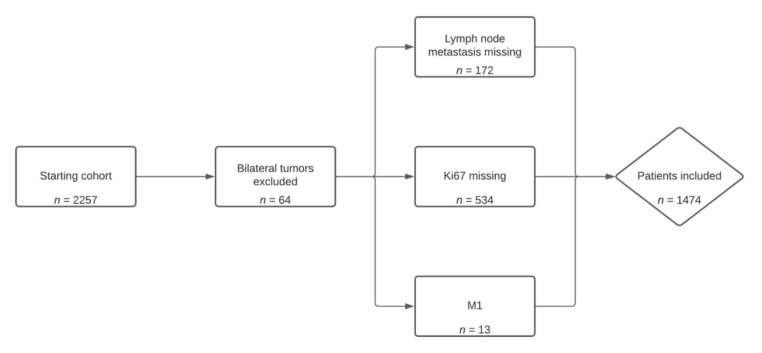
The database primarily comprised 2257 tumors. After the exclusion of 64 non-dominant bilateral tumors, this corresponded to 2129 patient cases including locoregional recurrence. However, in 172 patients, no information regarding the lymph node involvement could be found. In 534 patients, the Ki67 proliferation marker was not assessed, and 13 patients displayed metastatic disease. Hence, the study population consisted of 1474 patients.

**Figure 2 jpm-12-00382-f002:**
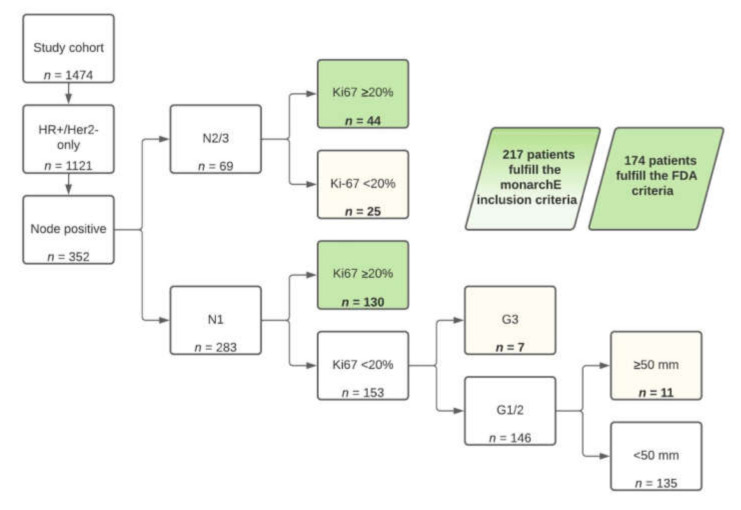
In total, 1474 patients comprised the study cohort, of whom 1121 had a HR+/Her2− breast cancer subtype. 352 of 1121 patients had involved lymph nodes. 69 patients displayed ≥4 pathologic lymph nodes (N2/3), of whom 44 exhibited tumors with a Ki67 ≥ 20% (green) and 25 a Ki67 < 20% (light yellow). A total of 283 patients showed 1–3 pathologic lymph nodes (N1), of whom 130 exhibited tumors with a Ki67 ≥ 20% (green). Of 153 patients with a Ki67 < 20% (light yellow), 7 displayed a high-grade tumor (G3), and 11 patients had a pathological tumor size > 50 mm (T3). Hence, 217 patients fulfilled the monarchE inclusion criteria (bold). A total of 174 patients met the FDA criteria for the approved use of abemaciclib, with a Ki67 ≥ 20% (green).

**Table 1 jpm-12-00382-t001:** Characteristics of HR+/Her2− patients.

	Number of Patients	Percentage
	1121	100%
**Age (years)**	59.79 ± 12.52	
**Menopausal status**		
Premenopausal	312	27.83
Postmenopausal	806	71.90
Male	3	0.27
**Histology**		
NST	850	75.83
ILC	193	17.22
Other	78	6.96
**Grading**		
1	156	13.92
2	735	65.57
3	228	20.34
n/a	2	0.18
**T-stage ***		
0	33	2.94
1	658	58.70
2	368	32.83
3	47	4.19
4	15	1.34
**N-stage ***		
0	769	68.60
1	283	25.25
2	48	4.28
3	21	1.87
**ER status**		
+	1115	99.46
−	6	0.54
**PR status**		
+	979	87.33
−	142	12.67
**Her2 status**		
+	0	0
−	1121	100
**Ki67**		
≥20%	454	40.50
<20%	667	59.50
**Chemotherapy**		
Neoadjuvant	62	5.53
Adjuvant	200	17.84
None	859	76.63

* T and N stages were assessed after surgery. NST, non-special type; ILC, invasive lobular carcinoma; ER, estrogen receptor; PR, progesterone receptor; Her2, human epidermal growth factor receptor 2; n/a, not available.

**Table 2 jpm-12-00382-t002:** Characteristics of patients fulfilling the monarchE inclusion criteria.

	Number of Patients	Percentage
	217	100%
**Age (years)**	60.23 ± 13.96	
**Menopausal status**		
Premenopausal	365	29.95
Postmenopausal	149	68.66
Male	3	1.38
**Histology**		
NST	175	80.65
ILC	34	15.67
Other	8	3.69
**Grading**		
1	2	0.92
2	121	55.76
3	94	43.32
n/a	0	0
**T-stage ***		
0	7	3.23
1	61	28.11
2	110	50.69
3	28	12.90
4	11	5.07
**N-stage ***		
0	2	0.92
1	148	68.20
2	46	21.20
3	21	9.68
**ER status**		
+	216	99.54
−	1	0.46
**PR status**		
+	182	83.87
−	35	16.13
**Her2 status**		
+	0	0
−	217	100
**Ki67**		
≥20%	174	80.18
<20%	43	19.82
**Chemotherapy**		
Neoadjuvant	26	11.98
Adjuvant	85	39.17
None	106	48.85

* T and N stages were assessed after surgery. NST, non-special type; ILC, invasive lobular carcinoma; ER, estrogen receptor; PR, progesterone receptor; Her2, human epidermal growth factor receptor 2; n/a, not available.

## Data Availability

The data presented in this study are available on request from the corresponding author. The data are not publicly available due to the privacy of sensitive patient data.

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
