# Peer review of "Feasibility of Adjuvant Treatment with Abemaciclib—Real-World Data from a Large German Breast Center"

_jpm, 2022, doi:10.3390/jpm12030382_

Round 1

Reviewer 1 Report

The current study is on a topic of relevance and general interest to the readers of the journal - This is a well written paper , some suggestions are

1) Although the authors mention the aim of the study, there does not seem to be a real novelty attached. Please mention a sentence or two about how does this study add to current knowledge? and does it challenge existing paradigms?

2) Please present some of the strengths and limitations of the study

3) Please summarize if you identified a range of challenges  opportunities and provide a framework for further proposed studies.

Author Response

Thank you very much for critically revising our publication. We improved our manuscript based on the suggestions of the reviewer. 

We could specify the aim of the study and stated what the study adds to current knowledge in the introduction.

"Presuming approval also by the European Medicine Agency (EMA), the aim of this study was to model how many patients with eBC would be eligible for abemaciclib treatment using the inclusion criteria of monarchE and clinical data from a large single-center university hospital in Germany. Since clinical trials are conducted under controlled conditions, this study characterizes the patient population benefiting of abemaciclib in a real-world setting."

Furthermore the suggestion to point out strengths and limitations of our study with focus on the implementation of further studies was very helpful. We review this point now adequately in a section of the discussion:

"This study could highlight the importance of characterizing the patient population that could benefit of abemaciclib in a real-world situation. The results of this study are of great importance for estimating the target group and thereby the cost-effectiveness of new therapeutic approaches. However, these findings should be evaluated carefully since we conducted a single-center analysis that might not reflect the standard of treatment decisions in Germany. Especially with regard to increasing treatment possibilities for breast cancer, not all putative drug combinations can be evaluated in ‘classical’ clinical trials. Therefore, multicentric registers are needed to precisely collect information about treatment process, clinicopathological risk factors, molecular data and patient outcome [5, 26]."

Reviewer 2 Report

The authors have presented a retrospective analysis of early Breast cancer patients treated at a tertiary cancer centre in Germany, over a period of two years to identify a number of patients who would be eligible for Abemaciclib treatment on applying monarchE study criteria. 

Although their finding is sync with several similar previous studies for various drugs and agents reflecting differences in data or numbers expected in trial settings versus real life. It would be great if they could add some discussion on how their findings would help insurance or funding agencies to imply coverage and how newer drugs clinical trials can be impacted.

It was nice to see the finding and are in limits of expectations. They should be helpful for healthcare planning and clinical trials.

Author Response

Thank you very much for your comments on our paper. We found the suggestions very helpful to improve our discussion with regard to strengths and limitations of our study. We now added the following section to the discussion part:

"

This study could highlight the importance of characterizing the patient population that could benefit of abemaciclib in a real-world situation. The results of this study are of great importance for estimating the target group and thereby the cost-effectiveness of new therapeutic approaches. However, these findings should be evaluated carefully since we conducted a single-center analysis that might not reflect the standard of treatment decisions in Germany. Especially with regard to increasing treatment possibilities for breast cancer, not all putative drug combinations can be evaluated in ‘classical’ clinical trials. Therefore, multicentric registers are needed to precisely collect information about treatment process, clinicopathological risk factors, molecular data and patient outcome [5, 26]."